# Camillo Golgi’s Impact on Malaria Studies

**DOI:** 10.3390/cells12172156

**Published:** 2023-08-28

**Authors:** Madalina Simoiu, Radu Codreanu, Antonio-Daniel Corlatescu, Andreeea Marilena Pauna, Suzana Elena Cilievici

**Affiliations:** 1Department of Parasitology, “Carol Davila” University of Medicine and Pharmacy, 020021 Bucharest, Romania; madalina.simoiu@umfcd.ro (M.S.); radu.codreanu@drd.umfcd.ro (R.C.); 2National Institute of Infectious Diseases “Prof. Dr. Matei Balș”, 021105 Bucharest, Romania; 3Department of Neurosurgery, “Carol Davila” University of Medicine and Pharmacy, 020021 Bucharest, Romania; antonio.corlatescu@gmail.com; 4Department of Epidemiology, “Carol Davila” University of Medicine and Pharmacy, 020021 Bucharest, Romania; andreea.pauna@umfcd.ro; 5Military Medicine Institute, 010919 Bucharest, Romania; 6Colentina Clinical Hospital, 021151 Bucharest, Romania

**Keywords:** Camillo Golgi, malaria research, histology, Golgi staining technique, cellular structures, *Plasmodium*, life cycle, parasite stages, transmission, mosquito vector

## Abstract

Camillo Golgi was an esteemed Italian physician and biologist who made major advances in malaria research between the late 19th and early 20th centuries. His groundbreaking contributions in histology, especially through the development of the Golgi staining technique, revolutionized our understanding of cell structures—including *Plasmodium* parasites—through visualization. Golgi staining also allowed researchers to observe its complex life cycle while documenting it. His careful observations of malaria led to the identification and characterization of its various stages, both asexual forms within human red blood cells, as well as sexual forms carried by mosquito vectors. Golgi’s research highlighted the key role mosquitoes play in malaria transmission. He demonstrated the presence of *Plasmodium* sporozoites within the salivary glands of infected mosquitoes, providing insight into its life cycle and the dynamics of parasite transmission. His comprehensive approach contributed significantly to our understanding of malaria as a systemic illness, leading to subsequent research efforts within this field. The Golgi Protein complex is often located within the cis-Golgi of blood parasite life cycles and mosquito stages, indicating its possible role in optimizing asexual development during blood stages. Furthermore, its expression can be conditionally repressed or its gene can be inactivated to optimize this potential role in improving its functionality for optimizing sexual development during blood stages. Camillo Golgi remains one of the leading lights of malaria research today. His innovative staining techniques, detailed observations, and insightful interpretations have laid the groundwork for subsequent discoveries and advancements in malaria studies. By deciphering intricate parasite life cycle interactions with hosts, his work has provided invaluable insights into malaria biology, pathogenesis, and epidemiology.

## 1. Introduction

Malaria remains one of the deadliest infectious diseases ever witnessed on earth, having left lasting scars on many of our ancestors’ lives throughout history. Thanks to remarkable research conducted since the nineteenth century, malaria has begun to become less frequent throughout Europe, and was finally eliminated during the second half of the XXth century. Many ancient European regions were *Plasmodium* parasite endemic zones; many historians even debate whether or not Rome fell because of this disease [1].

Over subsequent centuries, an abundance of Italian scientists, representing Italy as a whole, began researching the etiology of malaria. Camillo Golgi (1843–1926), among many notable Italian scientists who investigated this field, made significant contributions both in cell biology and malaria infection research, providing detailed accounts on Plasmodium infection processes within intraerythrocytic cycles [1,2]. Golgi pioneered groundbreaking work investigating temporal correlations between frequent fever attacks and increased malaria parasite counts within the human bloodstream. His contributions advanced not only the field, but also expanded on studies conducted by his predecessors who focused on uncovering the causes and propagation mechanisms of malaria within human blood. Camillo Golgi made his mark as an expert on malaria infection, becoming a pivotal figure both within this field and cell biology. His thorough descriptions of intraerythrocytic cycles responsible for quartan and tertian fever provided invaluable insights into the relationship between fever attacks and parasite proliferation within humans bloodstreams [3,4]. Golgi’s research is remarkable for many reasons. First and foremost, his remarkable contributions transcend neuroscience and cellular biology; rather, they span far beyond these fields. Golgi was deeply invested in infectious disease research, taking an intricate yet meticulous approach that combined careful observation with experimentation; this allowed him to uncover disease symptoms as well as their implications for progression, thus making his contributions particularly impactful regarding both scientific comprehension as well as medical issues at large.

Today, the parasite still thrives in subtropical and tropical regions in this vulnerable population including young children, pregnant women, and those involved in economic development activities [5]. Reports by the World Health Organization show an increase in cases of malaria over recent years, 247 million cases compared with 245 million in 2020 and an estimated total mortality figure of 619,000 deaths for 2021 (vs. 625,000 for 2020). At its height from 2020 to 2021, COVID-19 disruptions resulted in an unprecedented surge in malaria cases and deaths; approximately 13 million more malaria cases and 63,000 additional deaths were recorded globally during this time. Malaria remains endemic in the WHO African Region, accounting for 95% of cases and 96% of deaths between 2016 and 2021. Tragically, children under the age of five accounted for approximately 80% of deaths related to malaria, making control and prevention strategies vital in this vulnerable population [6].

## 2. Golgi Research Regarding Plasmodium

Golgi’s research provided much-needed insights into the complex biological cycle of malaria parasites living inside blood cells. His work shed light on their link to fever onset, providing vital data that paved the way for effective malaria therapies and prevention strategies. Simply put, Golgi provided invaluable knowledge of pathogenesis while proposing more efficient treatments and prevention methods [7]. At first glance, it may appear strange that Golgi’s name would be associated with diverse fields such as infectious diseases, specifically malaria, and neurosciences. However, this apparent dissonance can be better explained considering his era, when microbiology was just emerging as an academic discipline and Golgi was deeply immersed in its development. Golgi excelled in using the microscope, an essential instrument in positivist science. Pavia, in northern Italy, where Golgi worked proved to be an ideal place for microbiology research. Golgi began exploring this emerging field with a similar passion to that with which he pursued his study of nervous systems (Lazzaro Spallanzani’s and Agostino Bassi had pioneered it prior to Golgi). Golgi pursued both fields with equal determination [8].

Golgi recognized the microscope as a dynamic and effective research tool capable of producing significant findings in infectious disease research. His contributions were not limited to his own achievements alone, but included those made by exceptional disciples such as Adelchi Negri, Giovan Battista Grassi, Antonio Carini and Giuseppe Sanarelli among many others he mentored and inspired during their research careers. Golgi made many notable discoveries about Anthrax and *Strongyloides stercoralis*; however, he became particularly fascinated with malaria after Alphonse Laveran first identified parasites in a patient’s blood on 6 November 1880.

Noteworthy is Italy’s endemicity, with devastating health and social repercussions. Laveran’s findings were validated by Rome’s scientifically established malariological school under Ettore Marchiafava and Angelo Celli; however, many questions remained, such as why fevers were occurring during various seasons and an in various locations throughout Italy, why malaria produced different outcomes, and when quinine should be administered, leaving Camillo Golgi intrigued and inspired to study malaria at the Malariological School in Rome during a brief stint at Santo Spirito Hospital during September 1885.

Golgi returned to Pavia and immediately began conducting daily blood smear examinations of malaria patients at his clinical unit at the hospital. To his luck, only quartan malaria (“long fever”) existed in Pavia during the fall season, allowing for consistent observations [3].

Camillo Golgi presented his findings about *Plasmodium malariae’s* complete dynamic cycle and clinical observations of febrile attacks before the Royal Medical Academy in Turin on 18 November 1885. He used meticulous work with daily blood smears from malaria patients to unlock the secrets behind periodic fever, something which had long baffled scientists. Golgi found that febrile attacks could be predicted by watching for signs of maturation of segmented forms of the parasite that led to febrile attacks, known as quartan fever. He concluded that when observed together, mature and segmented forms could accurately forecast when another attack would come along, something no other scientist had accomplished prior to Golgi. His incredible ability to combine clinical observations with microscopic interpretation enabled him to make significant advances in our knowledge of malaria (Figure 1).

His research on malaria not only led to a greater understanding of the disease but also had practical ramifications. For instance, he noticed that the time when quinine was administered could influence its effectiveness against the parasites responsible for malaria; specifically, he noticed it being less effective after segmentation had already started; this finding became one of the earliest demonstrations of its schizonticidal properties, and this discovery long predated any formal pharmacokinetic studies [9].

Golgi made great strides toward understanding *P. malariae*, yet soon realized his observations were not applicable to all types of malaria. He observed the different forms that existed depending on where one lived; Rome and the surrounding countryside had its own type during spring, while intracellular parasites often appeared differently microscopic wise than traditional malaria parasites (i.e., *P. malariae*). To address this dilemma, Golgi observed cases in Pavia during the spring of 1886 and presented his findings on 5 June 1886 to the Medical Society of Pavia where he presented what has become known today as *Plasmodium vivax’s* complete human schizogonic cycle as described by Golgi himself [10].

Camillo Golgi made groundbreaking contributions to malaria research from 1885 to 1892, when microbiology made strides forward. Golgi built upon the work of Ettore Marchiafava and Angelo Celli to assess the reproductive cycles of *Plasmodium* parasites in human blood, noting the coincidental relationship between chills or fever and the release of merozoites into the circulation. He also demonstrated that intermittent fevers in adults were caused by two distinct protozoan organisms—*Plasmodium malariae* and *Plasmodium vivax*—contrary to Alphonse Laveran’s belief in only “*Oscillaria malariae*”, thus revolutionizing the malaria diagnosis and management strategies. His discoveries greatly advanced our knowledge and provided more effective approaches [11,12].

Golgi combined naturalistic, biological, and clinical-pathological perspectives in 1889 to develop an innovative method for the differential diagnosis of tertian and quartan fever, independent of clinical observation, creating valuable laboratory data which could then be applied directly for clinical diagnosis. Furthermore, his accomplishments included differentiating three distinct intermittent fever types as well as establishing optimal timing for quinine administration; all this greatly advanced our knowledge of malaria while setting new standards for its diagnosis and treatment [13].

He played an instrumental role in disproving Klebs and Tommasi Crudeli’s assertion that the “*Bacillus malariae*” found in Pontin Marshes south of Rome caused malaria, thus beginning an exciting era of research that saw Italian malariologists making significant contributions despite scientific rivalries and disagreements [12].

Fluorescence microscopy, biochemical assays and genetic approaches were employed by researchers to identify PfVPS51 protein that plays an integral part in Golgi-to-endosome trafficking pathways. Researchers demonstrated that PfVPS51 localizes to the Golgi apparatus and is essential for efficient protein sorting to endosomes. Researchers discovered that disabling PfVPS51 led to defects in parasite growth and development, further supporting its importance for *Plasmodium* survival and virulence. Their results contribute to our knowledge of intracellular trafficking mechanisms within *Plasmodium falciparum* cells and can also potentially be used as targets for antimalarial therapy treatments targeting this protein-sorting pathway [14,15] (Figure 2).

Malaria parasites display a less-developed Golgi apparatus compared to other eukaryotic cell types. Their Golgi apparatus is composed of dispersed, unstacked, cis- and trans-cisternae; despite its crucial role in *Plasmodium’s* secretory pathway, only a few Golgi resident proteins have been studied; Golgi Protein 1 (GP1) is yet to be investigated further, but we found evidence suggesting that it forms an association with previously uncharacterized transmembrane protein Golgi Protein 2 (GP2). Golgi Resident Protein 1’s complex with a previously uncharacterized transmembrane protein known as Golgi Protein 2 (GP2) but evidence was found regarding its involvement with Golgi Protein 2 [16,17].

The Golgi Protein complex is localized to the cis-Golgi during all stages of erythrocytic parasite life cycles as well as mosquito stages, with its expression being conditionally repressed or its gene inactivated, suggesting its significance at optimizing asexual development during blood stages. Analyses of parasite strains in which either expression was conditionally repressed and/or gene activation occurred have demonstrated this [18,19]. While not required at any stage in its life cycle, the optimization of blood stage development still has a vital role [16,17].

## 3. Plasmodium Development Study and Screening Methods

Malaria parasites have been studied using electron microscopy and live-cell microscopy to reveal their ultrastructure and dynamic processes, respectively, in real time. However, this research provides key insights into dynamic processes too small to be resolved using conventional live-cell microscopy; specifically, observations were made regarding P. falciparum organelle biogenesis as well as the organization of Plasmodium cell division around microtubule-organizing centers (MTOCs).

Ultrastructure expansion microscopy (U-ExM) should not be seen as a replacement for existing microscopy techniques; rather, it should be seen as an invaluable addition to our arsenal for studying malaria parasite cell biology. By filling some gaps left by electron and live-cell microscopy, respectively, U-ExM provides new insights into many parasite processes, the investigation of which would benefit greatly from U-ExM analysis; some examples have already been highlighted here, while many more await further exploration [20].

Merozoite invasion has been extensively studied using various microscopy techniques, but U-ExM could provide invaluable insight into how organelles associate and rearrange during this process. Furthermore, the rapid disassembly of inner membrane complex IMC organelles immediately following invasion may warrant U-ExM investigation; such rapid and dramatic rearrangements exceed live-cell microscopy resolution capabilities. U-ExM offers exciting opportunities for investigating the malaria parasite biology that was once inaccessible with traditional microscopy methods. Its implementation will complement existing techniques while offering insights into key aspects of the parasite’s life cycle and cellular processes [20,21].

CRISPR-Cas9 screens have generated an abundance of data, revealing multiple host genes influencing Plasmodium liver-stage parasite replication and development; some genes helped, while others hindered this process. Notably, certain proteins involved in cell trafficking, the immune response, and metabolic pathways were identified as having critical functions in ensuring successful liver stage parasite development; this finding offers exciting prospects for malaria control strategies. Targeting specific host genes identified in this research could provide innovative interventions that stop malaria’s progression, and, therefore, help to avoid symptoms of malaria. The CRISPR-Cas9 screen’s power lies in its ability to reveal complex interactions between Plasmodium parasites and host cells during liver stage malaria parasite development, providing insight into malaria biology. This groundbreaking research opens the way to revolutionary therapeutic approaches against malaria, from gene editing technology and host–parasite interactions to novel antimalarial agents. All these advances hold enormous promise in our collective effort to eliminate malaria globally [22].

*T. gondii* and *P. falciparum* share an evolutionary history and possess similar cellular processes, indicating that they should react similarly to inhibitors that affect their respective life stages. In an efficient image-based screening protocol using transgenic T. gondii expressing TgSORT-GFP and ROP1-mCherry, the ROP1-mCherry signal alone proved accurate for automated miniaturized drug screening; our aim was to identify drugs which disrupt subcellular localization, affecting parasite polarity morphology replication. Screening yielded compounds inhibiting p38-MAP kinase, such as Cpd2 (SB 203580) and Cpd3 (SB 203580 and SB 203580), respectively, known to reduce T. gondii growth [23].

The kinase inhibitors found during our screening may target some essential parasite kinases. While p38-MAP kinase inhibitors have shown some toxicity during previous clinical trials, other drugs unrelated to kinase inhibition identified in our research appear promising for consideration. Based on our findings, we propose that combination therapies with multiple effector mechanisms might be more successful than using just one or two with identical mechanisms of action, thus making the development of resistance against multiple effector mechanisms simultaneously more challenging [23].

Target identification and mechanism analysis are critical steps toward developing cross-species antiparasitic compounds with maximum efficacy. By connecting specific chemical scaffolds to their associated cellular phenotypes in two distinct parasites from evolutionarily related but distinct parasite families, a pathway for the comprehensive mechanistic analyses of interest can be opened. Biochemical and genetic approaches will play an integral part in targeting drug targets in the design of novel parasite inhibitors; such a comprehensive approach is key towards developing effective and targeted antiparasitic treatments [23].

## 4. Conclusions

Camillo Golgi’s diligent research allowed for an improved understanding of two prevalent plasmodia in Italy, and his findings provided invaluable assistance for Marchiafava and Celli who described, in 1892, the life cycle of the *P. falciparum* malaria parasite that causes the third type of malaria (malignant tertian malaria). Golgi’s era in the early 1890s witnessed the remarkable establishment of a taxonomy that has remained unchanged even after 130 years, specifically in relation to the genus Plasmodium and its four human species: *P. falciparum*, *P. vivax*, *P. malariae*, and *P. ovale*. Italian scientists of that time achieved an extraordinary feat in accurately describing and classifying these plasmodia, which is a rarity in the field of taxonomy, particularly when it comes to microbes. Typically, the initial observers and describers often make errors, but the Italians managed to get it right, leading to a taxonomy that has stood the test of time. Golgi’s work was completed prior to Romanowsky stain becoming available, further emphasizing his extraordinary scientific abilities. Camillo Golgi shared the Nobel Prize with Santiago Ramon y Cajal of Spain for their contributions to understanding the microstructure of the nervous system [24]. However, some have argued that his discovery of the Golgi apparatus in 1898, and his description of the malaria parasite life cycles in 1885 also made him a worthy candidate for consideration to receive the prize. Golgi also played a pivotal role in Adelchi Negri’s early 1900s campaign to combat malaria, with important documents about his contributions being held at Pavia’s Museum for History & Malariology (MUHP) [3,25,26].

## Figures and Tables

**Figure 1 cells-12-02156-f001:**
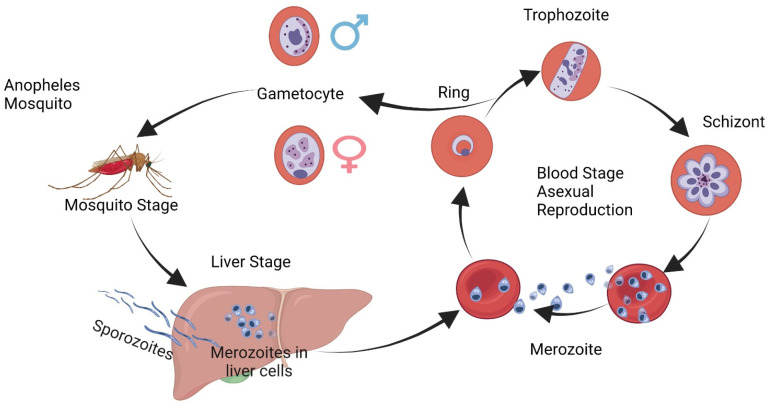
*Plasmodium malariae* Life Cycle: Illustration; stages of *P. malariae* include blood stages. Malaria parasite life cycles have many stages including sexual and asexual stages as well as intraerythrocytic development and transmission of *P. malariae* parasites.

**Figure 2 cells-12-02156-f002:**
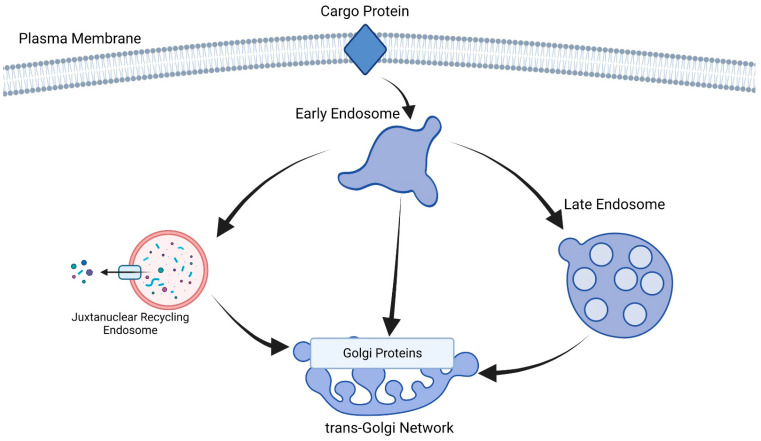
Protein cargoes ingested from the plasma membrane (PM) are transported back to the trans-Golgi network (TGN) via retrograde transport pathways that involve various endosomal compartments. Each protein cargo utilizes unique pathways within this endosomal system for retrograde transport. Retrograde transport equipment depends on which endosomal compartment it serves. Retrograde transport pathways involve specific protein cargoes utilizing specific endosomal compartments and using machinery that facilitates sorting, tethering, docking and fusion along their routes back to the TGN.

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
