# Peer review of "Camillo Golgi’s Impact on Malaria Studies"

_cells, 2023, doi:10.3390/cells12172156_

Round 1
Reviewer 1 Report
1. The provided proof had numerous errors of what appeared to be typesetting. Just two examples: a) lines 55-56, repetitive phrasing; b) line 58, repetitive wording. Many citations are not superscripted, and much else. This needs far more careful proofreading that it has yet received.
2. At line 69, the suggestion that endemic malaria persists through much of the tropics being attributable to a "failure to meet hygiene standards" is both overly simplistic and arguably offensive. The problem is not hygiene at all, but equitable access to diagnostic and treatment for the infection, along with natural habitats far more conducive to malaria transmission than that which occurred in Europe. Malaria of temperate latitudes was eliminated with relative ease (superior access to essential care and fragile transmission cycles), and supposedly superior hygiene was not involved in that success. In the specific case of Italy, sanitation did play a role, i.e., draining of the Pontine marshes and other aggressive measures by the fascist government of that era.
3. At line 76, malaria in Africa is not "epidemic" but "endemic".
4. Line 170-209. While the contemporary knowledge of Golgi apparatus in plasmodia is interesting, one cannot credit Camillo Golgi for the modern findings. The authors don't do so explicitly, but its presence in this article seems to do so implicitly. If there is a connection, it needs to be made more clear.
5. The authors shed no light on Golgi's role in the creation of what amounted to the taxonomy of the genus Plasmodium and the 3 known human plasmodia: P. falciparum, P. vivax, and P. malariae -- all of which were described by Italian scientists in Golgi's era (early 1890s). Remarkably, that taxonomy still stands 130 years later effectively unaltered. This is very rare in taxonomy of anything, but especially microbes -- the first observers/describers rarely get it right. The Italians did.
Author Response
Dear Reviewer,
We would like to thank you for your suggestions. We announce you that we addressed all your concerns.
Kind regards,
The collective of authors
Reviewer 2 Report
The authors describe the historical importance of Camillo Golgi’s contribution to malaria studies. Although more detailed accounts of Golgi’s biography and scientific discoveries are available from different sources (books, websites, published scientific articles), the historical aspect related to malaria discovery and research is of interest.
The objective of this opinion paper is not clear. The paper consists of two parts: (1) the contributions of Golgi to malaria research (lines 38-169) and (2) a short description about the Golgi protein complex (lines 170-211). The sources of the latter section are not mentioned (no reference cited concerning the work published? in 2014) although readers can surmise that the authors are talking about their own research. Moreover, the second part of this opinion paper clearly goes beyond the scope of the title of the paper “Camillo Golgi’s impact in malaria studies” because the authors are not referring to Golgi’s own work. The abstract does not even mention the contents of the second part of the paper. In my opinion, the second part of the paper should be deleted to keep the paper coherent and focused on Golgi’s works.
English needs a moderate amount of corrections, especially punctuations. All Latin names should be italicized, including Plasmodium and Strongyloides stercoralis.
MAJOR COMMENTS:
Abstract: The abstract refers to several aspects of Golgi’s work which are not discussed in the main text:
Line 23: “his studies illuminated mechanisms controlling parasite invasion, replication, transmission”
Lines 24-25: “Golgi’s research highlighted the key role mosquitoes play in malaria transmission”
Lines 25-26: “He demonstrated the presence of Plasmodium sporozoites within salivary glands of infected mosquitoes”
Line 27: “malaria’s impact on host tissues, such as the spleen and liver”
These aspects should be developed in the text (instead of GP1 and GP2).
Introduction, lines 38-79: These paragraphs are quite confusing. I suggest reorganization, for example:
1-start with: lines 39-40: “Malaria remains one of the deadliest...throughout history”
2-go to lines 68-79: Today, the parasite thrives mostly in subtropical and tropical regions... to “...in this vulnerable population.”*
Eliminate line 68 “Plasmodium had almost been eradicated from Europe.” This statement contradicts lines 42-43 “...malaria has begun to become less frequent throughout Europe before finally being eliminated during the second half of the XXth century”
3-go to lines 40-45: “Thanks to remarkable research since the XIXth century, malaria has begun to become less frequent throughout Europe before finally being eliminated during the second half of the XXth century.” Malaria had been malaria endemic in many regions in Europe for centuries. Many historians even debate whether Rome fell because of this disease in the Vth century.
4-lines 46-67
Introduction, lines 46-67: This paragraph is very confusing. There are several repetitions, as shown below. The entire paragraph should be rewritten for more clarity, avoiding repetitions.
Lines 49-50: “...included his detailed description of Plasmodium intraerythrocytic cycle that caused quartan and tertian fever attacks and multiplication within human blood”
and
Lines 58-59: “...including his work including extensive descriptions of intraerythrocytic cycles responsible for causes quartan and tertian fever”
Lines 51-52: “Golgi also pioneered temporal correlations between recurrent attacks of fever attacks and multiplication within human blood”
and
Lines 60-61: “Camillo Golgi also discovered its temporal relationships between attacks of fever attacks occurring before and parasite’s proliferation within human bloodstream.”
Lines 55-56 are incomprehensible: “as one such scientist, in the area as one scientist dedicated solely dedicated solely dedicated solely dedicated towards this field; one such scientist...”
Figure 1 and lines 114-124: This whole paragraph describes the clinical presentation of P. malariae infection. The figure caption says that it is an illustration of P. falciparum life cycle, but several images suggest P. malariae: the form of gametocytes (not banana-shaped), trophozoite that shows a band or equatorial form, a limited number of merozoites within the schizont. The figure 1 cpation should be revised to “Plasmodium malariae” instead of Plasmodium falciparum.
Plagiarism, Fig. 2 and its legend: This photograph and the accompanying legend were copied, word-for-word, from the following website: Marine Bentivoglia. Life and discoveries of Camillo Golgi. NobelPrize.org. Nobel Prize Outreach AB 2023. https://www.nobelprize.org/prizes/medicine/1906/golgi/article/. This figure must be deleted unless the authors obtained the authorization from the copyright owner.
Lines 170-181 and figure 3: This entire paragraph is based on the findings in “an article from 2014,” but the authors do not cite it. Please add the reference citation in the text as well as in the Reference list. Is it based on REF 14 (Coppola et al. 2012), which does not seem to be related to malaria? If required, please cite the source of Figure 3.
MINOR COMMENTS:
REF 4 is about Golgi’s works in neuroscience. Does it support the statement in lines 60-61 “Golgi also discovered its temporal relationship between attacks of fever attacks occurring before and parasite’s proliferation within human bloodstream (3,4).”?
REF 5 is outdated (published in 1997). Please cite a more recent reference.
Line 76, “malaria remains an epidemic in the WHO African region”: Malaria is generally endemic in Africa.
Lines 82-83, “Furthermore, his work shed light...”: I suggest: (delete “furthermore”) His work shed light on their link to fever onset, (comma) providing vital data that paved (not “could pave”) the way...
Line 85 “while creating more efficient treatments...”: while proposing more efficient treatments
Line 90, Pavia: I suggest – Pavia, in northern Italy, where Golgi worked, proved to be an ideal place...
Line 101-102: Alphonse Laveran first identified malaria parasites...November 6, 1880 (the year must be corrected).
Line 103: Italy’s malaria endemicity
Line 108: alongside Malariological School in Rome
Lines 114-115: Golgi presented his findings about the complete dynamic cycle and clinical observations of febrile attacks due to Plasmodium malariae... 1885. His meticulous work...unlocked the secrets...
Lines 115, 118, 119 “feverish attacks”: febrile attacks
Line 138 “weren’t”: his observations were not applicable
Line 141, “traditional malaria parasites”: The meaning is not very clear. I suggest – “traditional” malaria parasites (i.e. P. malariae)
Lines 146-147, “He made major advances during this timeframe”: The statement can be deleted. The meaning is the same as the preceding statement “Golgi made groundbreaking contributions to malaria research from 1885-1892...”
Lines 151-152: I suggest - contrary to Alphonse Laveran’s belief in the existence of a single malaria species coined “Oscillaria malariae”
Line 167: I would place quotation marks – “Bacillus malariae”
Line 192, “yet due to its crucial role”: despite its crucial role
Line 193: GP1 is yet to be investigated...
Line 195, “in our previous study”: Please provide the reference citation.
Lines 207-211, “we have identified a Golgi resident protein complex consisting of GP1 and GP2...during its asexual growth in blood stages”: These statements may be better placed in line 193, after “only few Golgi resident proteins have been studied.”
Line 215-216, “P. falciparum malaria parasite that causes third malignant malaria (tertian malign malaria)”: I suggest – P. falciparum malaria parasite that causes the third type of malaria (malignant tertian malaria)
moderate editing
Author Response
Dear Reviewer,
We would like to thank you for your suggestions. We closely read and addressed your concerns.
Kind regards,
The collective of authors
Round 2
Reviewer 2 Report
The authors took into consideration only the minor suggestions that I made in my first review. The historical part (lines 38-185) of the paper (Golgi’s life and scientific contributions) was considerably improved. However, almost none of my major concerns were addressed by the authors.
English of the revised text needs corrections. I repeat that all Latin names should be italicized (Plasmodium, P. falciparum, P. malariae, Plasmodium vivax, Oscillaria malariae, Strongyloides stercoralis, Bacillus malariae).
MAJOR COMMENTS:
Abstract: The abstract still does not reflect the entire content of this paper. The second part of the paper concerning Golgi proteins and trafficking (lines 186-227) is not mentioned in the abstract.
Plagiarism, Fig. 2 and its legend: This photograph and the accompanying legend were copied, word-for-word, from the following website: Marine Bentivoglia. Life and discoveries of Camillo Golgi. NobelPrize.org. Nobel Prize Outreach AB 2023. https://www.nobelprize.org/prizes/medicine/1906/golgi/article/. This figure must be deleted unless the authors obtained the authorization from the copyright owner.
Lines 189-197 and figure 3: This entire paragraph seems to be based on the findings reported in REF 14 (Coppola et al. 2012), which is not related to malaria. Unfortunately, however, I still do not understand the relation between REF 14, Figure 3, and Golgi/endosome trafficking pathways in malaria parasites. Please provide citations that support the statements about PfVPS51 protein and the source of Figure 3 (which is not REF 14).
Lines 206-214: This interesting paragraph about Golgi proteins 1 and 2 should be developed further in a separate paper because it lies completely outside the subject being treated in this paper (“Camillo Golgi’s impact in malaria studies”), and pertinent references should be cited. The authors say “We found evidence suggesting...” and “in our latest investigation...” but do not cite any references.
Unless the authors can support the statements made in lines 189-227, they should be deleted.
MINOR COMMENTS:
Repetitions should be avoided:
Lines 206 and 223: “Malaria parasites display a less developed Golgi apparatus” versus “Malaria parasites possess rudimentary Golgi apparatus”
Lines 220-222 and 225-227 “While not required at any stage in its life cycle, optimizing blood stage development plays a vital role” versus “while not essential, its presence contributes to optimal development during its asexual growth in blood stages.” This is a repetition of the same idea. One of them should be deleted.
Lines 51-54: “Plasmodium infection processes within intraerythrocytic cycles” and “its proliferation within human bloodstreams” are repetitions/redundant.
Line 60, “(1843-1926)”: This was already mentioned in line 48.
Lines 236-237, “his [Golgi’s] discovery...and their [as it stands, “their” seems to refer to Golgi and Ramon y Cajal, which is not the case] description of malaria parasite life cycles. This should be clarified.
English of the revised text needs corrections. I repeat that all Latin names should be italicized (Plasmodium, P. falciparum, P. malariae, Plasmodium vivax, Oscillaria malariae, Strongyloides stercoralis, Bacillus malariae).
Author Response
Dear Reviewer,
We have read your comments addressed the following:
- We have updated the abstract, including the second part of the paper
- Fig.2 has been deleted, even though it was the original figure of camillo golgi and was cited accordingly, we have decided to remove it.
- We have added a reference for the lines 189-197 and figure 3
- We have removed the part about golgi proteins 1 and 2
- We have addressed the minor comments
Thank you for your feedback
Kind regards,
The collective of authors